# A Descriptive Study of 103 Primary Cutaneous B-Cell Lymphomas: Clinical and Pathological Characteristics and Treatment from the Spanish Lymphoma Oncology Group (GOTEL)

**DOI:** 10.3390/cancers16051034

**Published:** 2024-03-03

**Authors:** Natividad Martínez-Banaclocha, Francisca Martínez-Madueño, Berta Caballé, Joan Badia, Mar Blanes, David Aguiar Bujanda, Virginia Calvo, Jose Gómez Codina, Cristina Quero Blanco, Pablo Espinosa, Javier Lavernia, Francisco Ramón García Arroyo, María Guirado Risueño, Cristina Llorca, Raquel Cumeras, Mariano Provencio Pulla, Josep Gumà

**Affiliations:** 1Medical Oncology Department, Hospital General Universitario Dr. Balmis, Institute for Health and Biomedical Research (ISABIAL), 03010 Alicante, Spain; 2Southern Catalonia Institute of Oncology, Hospital Universitari Sant Joan de Reus, IISPV-URV-CERCA, 43204 Reus, Spain; francisca.martinez@salutsant-joan.cat (F.M.-M.); berta.caballe@salutsantjoan.cat (B.C.); joan.badia@iispv.cat (J.B.); raquel.cumeras@iispv.cat (R.C.); josep.guma@salutsantjoan.cat (J.G.); 3Dermatology Department, Hospital General Universitario Dr. Balmis, Institute for Health and Biomedical Research (ISABIAL), 03010 Alicante, Spain; marblanes76@gmail.com; 4Medical Oncology Department, Hospital Universitario de Gran Canaria Dr. Negrin, 35010 Las Palmas de Gran Canaria, Spain; dagubuj@gobiemodecanarias.org; 5Medical Oncology Department, Hospital Universitario Puerta de Hierro-Majadahonda, 28222 Madrid, Spain; vircalvo@hotmail.com (V.C.); mprovenciop@gmail.com (M.P.P.); 6Medical Oncology Department, Hospital Universitari i Politècnic La Fe, 46026 Valencia, Spain; jgcodina@outlook.es; 7Medical Oncology Department, Hospital Universitario Virgen de la Victoria, 29010 Málaga, Spain; cquerob@gmail.com; 8Dermatology Department, Hospital Infanta Cristina, 28981 Parla, Spain; espinosapab@gmail.com; 9Medical Oncology Department, Fundación Instituto Valenciano de Oncología (I.V.O.), 46009 Valencia, Spain; jlavernia@fivo.org; 10Medical Oncology Department, Complexo Hospitalario Universitario de Pontevedra, 36071 Pontevedra, Spain; frgarroyo@seom.org; 11Medical Oncology Department, Hospital General Universitario de Elche, 03203 Alicante, Spain; mariaspalux@hotmail.com; 12Medical Oncology Department, Hospital General Universitario de Elda, 03600 Alicante, Spain; cllorcaf@seom.org

**Keywords:** cutaneous B lymphoma, skin, surgery, radiotherapy, chemotherapy, rituximab

## Abstract

**Simple Summary:**

Primary cutaneous B-cell lymphomas account for approximately 25% of all cutaneous lymphomas and 2% of all non-Hodgkin’s lymphomas. Three main entities are recognized: primary cutaneous marginal zone lymphoma, primary cutaneous follicle centre lymphoma, and primary cutaneous diffuse large B-cell lymphoma, leg type. In this article, the Spanish Lymphoma Oncology Group (GOTEL) presents a series of 103 primary cutaneous B-cell lymphomas.

**Abstract:**

Primary cutaneous B-cell lymphomas (PCBCLs) are B-cell lymphomas that can occur in the skin without evidence of extracutaneous involvement. The 2005 WHO/EORTC classification of cutaneous lymphomas and its 2018 update have distinguished three main categories based on clinicopathological, immunohistochemical, and genetic characteristics: primary cutaneous marginal zone lymphoma (PCMZL), primary cutaneous follicle centre lymphoma (PCFCL), and primary cutaneous diffuse large B-cell lymphoma, leg type (PCDLBCL-LT). PCMZL and PCFCL are clinically indolent, while PCDLBCL-LT is an aggressive lymphoma. Due to its low incidence and lack of prospective studies, it is difficult to establish a standard treatment for each subgroup. The objective of our study was to describe the clinical and pathological characteristics of 103 patients with cutaneous B-cell lymphoma from 12 centres belonging to the Spanish Lymphoma Oncology Group. The median age was 53 years (40–65). According to skin extension, 62% had single-site lymphoma, 17% had regional lymphoma, and 20% had multifocal lymphoma. Histology: 66% had PCMZL, 26% had PCFCL, and 8% had PCDLBCL-LT. Twenty-three percent of the patients were treated exclusively with surgery, 26% with radiotherapy only, 21% with surgery plus radiotherapy, 10% with polychemotherapy, and 5% with rituximab monotherapy. Overall, 96% of patients achieved a complete response, and 44% subsequently relapsed, most of them relapsing either locally or regionally. The 10-year OS was 94.5% for the entire cohort, 98% for the PCMZL cohort, 95% for the PCFCL cohort, and 85.7% for the PCDLBCL-LT cohort. Our data are comparable to those of other published series, except for the high frequency of PCMZL. The expected heterogeneity in therapeutic management has been observed.

## 1. Introduction

Approximately 25% of non-Hodgkin lymphomas (NHLs) originate in extralymphatic organs. The main extranodal location of NHL is the gastrointestinal tract, followed by the skin. Cutaneous lymphomas are mostly of the T lineage, and only 25–30% of them are B-cell lymphomas. Approximately 2% of NHLs are estimated to be primary cutaneous B-cell lymphomas (PCBCLs). The exact incidence of this group of lymphomas is difficult to determine, but population-based data from the US Surveillance, Epidemiology, and End Results (SEER) program show an adjusted incidence of 0.31 cases per 100,000 person-years (0.4 for men and 0.23 for women), with an M/F ratio of 1.76 [1]. Differences in relative frequency can be found between the different subtypes of PCBCL according to different authors [1,2,3]. This disparity may be due to the classification used in each series of patients. However, it is possible that there are real differences depending on the geographical area from which the patients were studied. The first modern classification specifically designed for cutaneous lymphomas was created from the consensus of the European Organization for Research and Treatment of Cancer (EORTC) Cutaneous Lymphoma Project Group [4]. This classification considers clinical and immunophenotypic characteristics, in addition to histological aspects, and classifies cutaneous B-cell lymphomas into three categories: indolent (follicle centre cell lymphoma and marginal zone B-cell lymphoma/immunocytoma), intermediate (large B-cell lymphoma of the leg), and provisional (intravascular large B-cell lymphoma and plasmacytoma). In 2001, the subsequent World Health Organization (WHO) classification of haematolymphoid tumours did not specifically classify primary cutaneous B-lymphomas. It was not until 2005 that the WHO-EORTC consensus classification of cutaneous lymphomas was published [5]. This classification system has remained practically unchanged until recently (only updated in 2018 [6]) and includes three main cutaneous B-lymphomas, namely primary cutaneous marginal zone B-cell lymphoma (PCMZL), primary cutaneous follicle centre lymphoma (PCFCL), and primary cutaneous diffuse large B-cell lymphoma, leg type (PCDLBCL-LT), revealing that PCMZL and PCFCL exhibit indolent behaviour, while PCDLBCL-LT has an aggressive clinical course. Intravascular large B-cell lymphoma is a rare entity in which neoplastic cells occupy the blood vessels and generally affect the CNS; however, cases with exclusively skin involvement have been described. According to the 2018 WHO-EORTC update, EBV^+^ mucocutaneous ulcers were considered provisional entities that affect immunosuppressed patients and can infiltrate the skin and oropharyngeal and gastrointestinal mucosa.

Considering only the histological characteristics of PCBCL, it is understandable how some of these tumours have been classified differently in clinical practice over the last 20 or 30 years. PCMZL shows an infiltrate with a nodular or diffuse pattern of small B cells: marginal zone cells (centrocite-like), lymphoplasmacytoid cells, and plasma cells. PCFCL is a centrofollicular cell neoplasm that comprises a mixture of medium-large centrocytes and centroblasts; it adopts a diffuse pattern in 65% of cases, a nodular and diffuse pattern in 30%, and a pure nodular pattern in only 5% of cases. PCDLBCL-LTs exhibit a diffuse pattern composed almost exclusively of centroblasts and immunoblasts. In the case of PCMZL, confusion is rare, and only some lymphomas formerly classified as cutaneous plasmacytomas or immunocytomas due to their predominance of plasma cells are currently classified as PCMZL. The probability of confusion is greater between the PCFCL and the PCDLBCL-LT. It is not uncommon for diffuse proliferation of predominantly large B cells, corresponding to the PCCFL, to be classified as diffuse large B-cell lymphoma and confused with PCDLBCL-LT if it is located in the skin. Because of this misdiagnosis, especially if the criteria of the WHO classifications of haematolymphioid tumours are used, a large number of previously classified cutaneous diffuse large B-cell lymphomas cannot be considered as such but are most likely PCFCLs, which have different clinical behaviours and, therefore, a different therapeutic approach [3]. Thus, the relative percentages of the three main histological variants of PCBCL vary considerably among the different published series: PCMZL (25–35%), PCCFL (30–50%), and PCDLBCL-LT (9–16%) [1,3].

For the correct diagnosis and staging of PCBCL, a complete clinical history and physical examination, including an examination of the patient’s entire skin, are needed. In addition, an adequate sample should be obtained by incisional or excisional biopsy or by means of a punch with a minimum diameter of 4 mm. Complete blood count and biochemical analysis (LDH, etc.), and serology for HIV, hepatitis, and Borrelia burgdorferi should be performed. To rule out distant dissemination or lymph node involvement, imaging with CT is recommended for indolent subtypes, whereas PET-CT is recommended for the most aggressive subtypes.

If any suspicious lymph node is found, histological confirmation is recommended. Bone marrow biopsy is recommended for patients with PCDLBCL-LT if the PET-CT result is unclear or if there is a high suspicion of infiltration due to cytopenia. The Ann Arbor classification should not be used to stage primary cutaneous lymphomas. In 2007, the TNM classification system for primary cutaneous lymphomas other than Mycosis Fungoides and Sézary syndrome was published [Kim YH; Blood 2007]. This was a consensus proposal between the International Society for Cutaneous Lymphomas (ISCL) and the Cutaneous Lymphoma Task Force of the European Organization for Research and Treatment of Cancer (EORTC). This TNM system aims to document the anatomical extent of lymphoma in the skin, but it has no prognostic value, at least for indolent cutaneous lymphomas. However, this approach is currently the recommended system for staging cutaneous B lymphomas.

A basic immunohistochemical panel for the pathological diagnosis of PCBCL, including CD3, CD20, CD10, BCL2, BCL6, IRF4/MUM1, and CD21, should be available; this panel can be broadened in some doubtful cases [7].

PCFCL accounts for approximately 50% of patients, with a median age at diagnosis of 55 years and a male predominance (1.5:1). The most common lesions are macules, papules, plaques, or purplish erythaematous tumours located mostly on the head and trunk. Patients with such lesions tend to experience recurrence in areas other than those previously treated (especially if a local treatment such as surgery or radiotherapy has been used), and there is usually no extracutaneous dissemination. The five-year disease-specific survival rate is approximately 95% [1,2,8]. As previously discussed, these tumours often exhibit a follicular and diffuse pattern of growth that can make them difficult to distinguish from PCDLBCL-LTs; in some cases, FISH or massive sequencing techniques are required for confirmation. The cells expressed the following B-cell lineage and germinal centre markers: CD19+, CD20+, CD79a+, PAX5+, IgM−, IgD−, CD5−, CD10−, Bcl-6+, IRF4/MUM1−, and FOXP1−. MYC expression is negative, and translocation (14;18) is extremely rare in PCFCL, which differentiates it from its nodal counterpart [9,10].

PCMZL is the second most common PCBCL after PCFCL in the majority of published series. The age range of presentation is broader than that for PCFCL, affecting middle-aged adults (50–60), children, and young adults. [11]. Clinically, it manifests as a single lesion or as multifocal reddish or violaceous papules, plaques, or nodules. The most frequently involved sites are the trunk and extremities, although other areas may also be affected. Like with PCFCL, it tends to recur on the skin, while extracutaneous spread is rare. This subtype has a five-year overall survival rate of approximately 99% and has now been renamed primary cutaneous marginal zone lymphoproliferative disorder (PCMZLD) because of its completely indolent biological behaviour [3,12]. It has been associated with several infectious agents, such as Borrelia burgdorferi (especially in Europe), *Helicobacter pylori*, vaccines, arthropod bites, traumatic lesions, tattoos, etc., but the aetiology of this disease has largely not been determined. Histologically, the infiltrate consisted of scattered follicles with small lymphocytes, small centrocyte-like B cells, and lymphoplasmacytoid and plasma cells. The immunophenotype is positive for CD20, CD79a, and bcl-2 and negative for CD5, CD10, and bcl-6, which differentiates it from PCFCL [13,14,15,16,17,18]. Depending on immunoglobulin expression, two subtypes of PCMZL have been differentiated: those expressing IgG, IgA, and IgE heavy chains with a better prognosis and those with an IgM subtype related to a higher frequency of extracutaneous involvement [19].

PCDLBCL-LT is the most aggressive variant of PCBCL and accounts for 15–20% of cases. The median age at diagnosis is 70 years, patients are predominantly female (2:1), and extracutaneous involvement is found in up to 35% of patients. Patients present with solitary or multiple nodules on their legs (although other skin locations are not uncommon) that are red or bluish in colour and can cause pain and necrosis due to rapid growth and ulceration. PCDLBCL-LT is the cutaneous B-cell lymphoma with the lowest survival (50–70% at 5 years) [7,20]. Histologically, dermal and subcutaneous infiltration by centroblasts and immunoblasts is observed, with a diffuse pattern and a high mitotic index. Neoplastic cells are positive for CD19, CD20, CD22, CD79a, IgM, Bcl-6, MUM1/IRF4, FOXP1, and c-myc, but Bcl-6 and CD10 exhibit variable expression. Therefore, activated B cells generally exhibit a double-expression phenotype, although they are rarely double-hit lymphomas. Mutations in the NFkB pathway (L265P MYD88, TNFAIP3/A20, CD79B, and CARD11) are often present. The MYD88 mutation is diagnostic for this subgroup, as it is absent in other primary cutaneous B-cell lymphomas. The rearrangement of cMYC and inactivation of CDKN2a by promoter deletion or hypermethylation are involved in the pathophysiology of PCDLBCL-LT and are associated with poor prognosis [21]. Rearrangements of the Bcl2 gene are rare. Up to 40% of PCDLBCL-LTs may have alterations in PD-L1/PD-L2, leading to an immune-invasive microenvironment that favours tumour progression and could have therapeutic implications [22].

The low incidence of PCBCL has led to great variability in therapeutic management, often extrapolated from patients’ lymph node counterparts. For indolent PCBCLs such as PCFCL and PCMZL, surgical excision or local radiation therapy is the most common approach for treating single lesions. For patients with multiple lesions in a limited area, radiation therapy is the preferred treatment if the lesion can be encompassed in a single radiation field. Doses of 30–45 Gy are usually used, with margins of 1–2 cm. Treatment can be delivered using electrons in superficial lesions or with high-energy photons in thicker tumours. This approach results in a complete response rate of approximately 99%, although approximately half of patients will experience recurrence, usually outside the irradiation field [23,24]. Localized cutaneous recurrences can be retreated by surgical excision or with radiation therapy if they occur at previously unaffected sites. When there is a multifocal disease that cannot be covered in the radiation field, treatment should follow a philosophy similar to that of advanced indolent nodal lymphoma: judicious use of systemic or local therapies with palliative intent, considering that these patients have a long life expectancy, and quality of life is of utmost importance. Therefore, therapeutic abstention in asymptomatic patients is prioritized. Local radiation therapy (even at low doses such as 4 Gy) can be used for symptomatic lesions [25]. Systemic treatment with oral chlorambucil or rituximab alone can achieve significant responses. Polychemotherapy is rarely necessary, and although it achieves good response rates, recurrences are approximately 50%. With intralesional rituximab, responses greater than 80%, which can last for months or years, can be obtained. In addition, better cosmetic results are obtained via surgery or radiotherapy than via surgery or radiotherapy, although long-term results are lacking [26,27,28]. In PCMZL, especially if the infection is associated with positive serology for Borrelia, antibiotic treatment is worth considering.

PCDLBCL-LT should be treated with the same strategy used for primary nodal DLBCL. Combined treatment with three to six cycles of R-CHOP and involved-field radiation is recommended for patients in localized stages. In advanced stages, six cycles of R-CHOP could be considered the standard treatment, analogous to nodal DLBCL. In patients in whom a complete response is not achieved or in those who respond slowly to chemoimmunotherapy, the addition of radiotherapy should be considered.

Due to the diversity of therapeutic options, the choice of treatment should be based on the clinical stage, age, and general condition of the patient through a multidisciplinary approach by a team experienced in cutaneous lymphomas, integrated by dermatologists, haemato-oncologists, and radiotherapists.

Here, the Spanish Lymphoma Oncology Group (GOTEL) report the clinical, pathological, and outcome results of a retrospective series of 103 primary cutaneous B-cell lymphomas.

## 2. Materials and Methods

The medical records of patients with primary cutaneous B-cell lymphoma diagnosed between January 2002 and January 2022 at the participating centres were reviewed. Informed consent was obtained from the living patients. The data were compiled through a database designed for this purpose, in which information on sociodemographic variables, skin location of lymphoma, stage, histological classification, treatment, and outcome was collected. The type of imaging test performed on each patient was not recorded, since in all cases, it had to have been confirmed prior to inclusion in the study that they were PCBCL, so that the extension study had to be complete and conform to the WHO definition of PCBCL. In any case, all patients underwent a thoracic and abdominal CT scan and a complete physical examination to rule out lymphoma of lymph node origin

All the statistical analyses were performed with R software (version 4.2.2). Descriptive statistics are represented as n and % for categorical variables and as the median and range for numerical variables that did not follow a Gaussian distribution. Descriptive tables were generated with the help of the gtsummary R library (v.1.7.0). For univariate statistics, parametric assumptions were assessed using the Shapiro‒Wilk test, Q‒Q test and density plots; if the parametric assumptions were not met, nonparametric tests were chosen. Bivariate inferential analysis with Pearson’s chi-square test of independence was used to check for possible associations between categorical variables, such as sex, history, location of the lymphoma, histological classification, type of lymphoma, and stage. PFS was defined as the length of time between the date of diagnosis and the date of the event and a relapse or, in the absence of the event, the date of last contact. Similarly, overall survival (OS) was defined as the length of time between diagnosis and death or right-censored at last contact in the absence of an event. Disease-specific survival (DSS) was defined as the length of time between diagnosis and the event, death by lymphoma, or in the absence of the event, the date of death by other causes or the last contact date. Survival analysis was subsequently carried out using the nonparametric Kaplan‒Meier method and the log-rank test to compare survival curves according to the defined groups as well as the R libraries survival (v.3.5-0) and survminer (v.0.4.9). In the analyses, the results were considered to be statistically significant at a *p* value < 0.05.

## 3. Results

Between January 2002 and January 2022, one hundred and three patients were included in this retrospective study. The main clinical characteristics are detailed in Table 1.

### 3.1. Epidemiology

More than 70% of cases occurred in patients older than 40 years, regardless of histological subtype, with a median age of 53 years (11–85). By subtype, the median ages were 51 (11–82), 56 (24–77), and 66 (40–85) years for PCMZL, PCFCL, PCDLBCL, and LT, respectively (Appendix A). Overall, the patients presented a good performance status, with an Eastern Cooperative Oncology Group (ECOG) score of 0–1 in 96% of the patients. There were no sex differences in the whole cohort; however, stratifying by histology, there was a male predominance in the PCMZ and PCDLBCL-LT subgroups: 53 vs. 47% and 62 vs. 38%, respectively. In contrast, in the PCFCL group, the female sex was more common (33 vs. 67%).

For the possible risk factors related to PCBCL, we did not find any factor that could be related to the aetiology, such as tattoos or Borrelia burgdorferi infection. It should be noted that in most of the cases, these epidemiological data were not included in the medical records reviewed, and of those for which data were available, only 7.4% of the PCBCL (23.5% of the PCMZL) had serology for Borrelia burgdorferi, with only one positive case (Appendix A).

### 3.2. Histology

The most common histology was PCMZL (66%), followed by PCFCL (26%) and PCDLBCL-LT (8%). Information on the immunophenotype related to the cell of origin was available for seven of the eight PCDLBCL-LT patients. Four of these patients had activated DLBCL, and three had germinal centre DLBCL (Figure 1) (Appendix A).

### 3.3. Location and Extension

Up to 98% of the patients were asymptomatic and diagnosed with Ann Arbor localized stage I-II disease, which was similar for all three subtypes: 98% with PCMZL, 100% with PCFCL, and 87% with PCDLBCL-LT. Only two patients were diagnosed with stage IV disease at diagnosis—one with PCMZL and one with PCDLBCL-LT (2%). The most common location was the trunk (54%), followed by the head and neck (23%) and the extremities (19%). Four percent (4/103) of patients developed cutaneous disseminated disease in several locations, all of whom had the two low-grade subtypes (three PCMZL; one PCFCL). Curiously, the most common location of PCFCL was not the scalp (30%) but the trunk (52%), and the most common location of PCFCL was not the legs (12%) but the head and neck (38%). Most of the tumours presented as single location (62%) or regional (17%), with 20% of the tumours being multifocal. In one patient with PCMZL, the extension could not be found in the medical history records (1%) (Table 1).

Multifocality was more frequent in the PCFCL subgroup (26%) than in the PCMZL and PCDLBCL-LT subgroups (19% and 12%, respectively) (Figure 2).

### 3.4. Types of Treatments

Radiotherapy (RT) was the most commonly used local treatment and was administered to 55% of the patients. Surgery (S) was used globally in up to 50% of patients. Both local treatments were used in some patients in combination with initial systemic treatment (ST) (RT + ST, 7%; S + ST, 5%; S + RT + ST, 1%). Exclusive local treatments were used in 70% of the patients; these treatments consisted of surgery (23%), radiotherapy (26%), or both (21%) (Table 2). Forty-two percent of the patients treated with RT received doses of 36–40 Gy (19/37 PCMZL; 5/16 PCFCL). Among patients for whom radiotherapy dose information was available, 17% received 30–35 Gy, 67% received 36–40 Gy, and 16% received more than 40 Gy (Table 2).

Only 25% of the patients underwent some type of systemic treatment associated or not associated with radiotherapy or surgery, mostly corresponding to PCDLBCL-LT (62%). The most commonly used systemic treatment was R-CHOP (8/26), followed by rituximab monotherapy (7/26) and CHOP (4/26). Rituximab monotherapy was used only for PCMZL (57%) and for PCFCL (42%) but not for PCDLBCL-LT. In our series, only three patients received intralesional treatment—two (one PCMZL and one PCFCL) with rituximab and another (PCFCL) with corticosteroids (Table 3).

### 3.5. Response, Recurrence, and Survival

The tumour response was complete in 96% of the patients in the global cohort, whereas it was lower in the leg-type lymphoma subgroup, in which 12% of patients did not respond to the treatment administered (CR: PCMZL 99%; PCFCL 93%; PCDLBCL-LT 88%) (Appendix A). OS did not reach the median in patients who achieved a complete response (CR) after the first treatment compared to those who did not achieve a CR after 60 months. As expected, patients with localized disease had better overall survival than those with advanced disease (*p* < 0.0001). Patients treated with surgery plus radiotherapy had a nonsignificantly greater disease-free survival (DFS) but not a longer overall survival (Appendix A). No significant differences were found in survival attributable to sex or age, but significant differences were found in relation to the PS; patients with a PS ranging from 0–1 had better survival than those with a PS2 (*p* < 0.001). A total of 43.6% (45/103) of patients had some type of recurrence. Among the 43.6% of patients who presented with some type of recurrence (45/103), 49% had local relapse, 20% had regional relapse, and 22% had distant relapse. In 9% of the patients, the type of recurrence was not recorded in the medical records. Seventy-eight percent of patients had between one and three recurrences, and of these, 62% had local recurrences. In contrast, only 8/45 patients had greater than three recurrences, with a greater number of regional and distant recurrences (76%). Patients with multifocal tumours at diagnosis had higher rates of global and distant recurrence than patients with nonmultifocal tumours did (69 vs. 37% and 33 vs. 18%, respectively). The median progression-free survival (PFS) rate in patients with multifocal tumours was lower than that in patients with nonmultifocal tumours (39 vs. 208 m, *p* = 0.014), but we found no significant difference in overall survival (OS, *p* = 0.68) (Figure 3D,F). The median PFS was 7.75 years in the complete group, with an overall survival rate of 10 years of 94.5% and a slight decrease in the leg-type subgroup (85.7%). Disease-specific survival at 10 years (DSS) was 97.2% in the whole cohort, while the PCDLBCL-LT subgroup had less specific survival (85.7%) (Figure 3, Appendix A). Only 3% of the deaths in the overall cohort were attributable to lymphoma.

## 4. Discussion

In our study, the median age for the different subtypes did not differ from that already known in the different publications, as a pathology with more frequent presentation at maturity. A subgroup of patients had PCDLBCL-LTs presenting at older ages (65% > 60 y), which, apart from its aggressive nature, may contribute to the worse survival of these patients given that older people also tend to have more comorbidities and poorer tolerance to polychemotherapy. There was no sex predominance in the complete group of PCBCL patients. Male sex predominated in the PCMZL (36/68) and PCDLBCL-LT (5/8) subgroups. In contrast, in the PCFCL, more cases were detected in females (9/27). These data contradict most published data where there is a slight predominance of males in the PCFCL subgroup and a greater number of female patients in the PCDLBCL-LT subtype. These discrepancies may be attributed to selection bias due to the retrospective nature of the study and the limited sample size, especially in the PCDLBCL-LT subgroup (n = 8). However, in a series published in 2022 with more than 4000 cases of primary and secondary cutaneous B-cell lymphomas, no significant differences were found with respect to sex in either the case of PCMZL or PCDLBCL-LT subtypes, so perhaps this issue remains an area for further study [29].

The vast majority of patients (98%) were diagnosed at localized stages (I-II), which corresponds to most of the studies published with PCBCLs, although one of the weaknesses of our study is that we do not have the data from the radiological scans performed on each of the patients for staging and, given the long period of data collection, these studies may have been changing over the years, with a greater number of PET-CT scans in recent years compared to the first ones, and this may have implications for the actual staging of these patients, which could have led to a greater number of advanced stages.

We did not find a relationship between the presence of Borrelia burgdorferi and the presence of PCBCL. Although Borrelia burgdorferi infection appears to be significantly associated with PCMZL, this association seems to be demonstrated only in endemic areas such as Australia, Scotland, and northeastern Italy. In our series, only 7.4% of PCBCLs and 23.5% of PCMZLs had borrelia serology, and only one PCMZL was positive. We have no information on the detection methods used (PCR, Southern blot, sequencing, etc.). It is unclear whether there is a direct benefit of antibiotic treatment, as scientific evidence is scarce, although antibiotic treatment may be considered a first therapeutic option in patients with serology-positive PCBCL and localized disease. However, the costs of diagnostic tests for detecting the presence of Borrelia burgdorferi and the use of antibiotics seem to be justified only in endemic areas. [30]

The most common histology was PCMZL, which was found in 66% of the patients, followed by PCFCL in 26% and leg-type DLBCL in 8%. This finding differs from the published data and may be related mostly to selection bias due to the retrospective nature of the study. However, in a recent publication of the prospective registry of cutaneous lymphomas from 2016 of the Spanish Society of Dermatology and Venerology, a higher percentage of PCMZL cases than of PCFCL cases was also observed, which may indicate a change in their incidence [31]. Another possible explanation is the application of immunohistochemical studies that were scarce before 2002 and that were implemented in successive years. A better identification of this disease in borderline cases with B-cell-rich pseudolymphomas could also explain these changes [32]. Among the eight patients with DLBCL-LT, seven had a confirmed immunophenotype, four had non-GCB, and three had GCB according to the Hans algorithm, which is in accordance with published data. However, we have no data showing that genomic sequencing was performed for any of our patients. Undoubtedly, the development of technology, the application of diagnostic criteria with improved immunohistochemistry, and new genomic sequencing techniques will lead to better classification of PCBCL [33]. One of the weaknesses of this study is that the biopsies were not reviewed at the central level, so there may also be irregularities in the histological classification of some of the patients [34].

For the locations, in the case of PCMZL, the trunk was the most common location, as described in the literature (40/68); however, in both the PCFCL (8/27) and the PCDLBCL-LT (1/8), the scalp and legs were not the most common locations. In contrast, in follicular lymphoma, the most common location was the trunk (52%), as in the case of marginal lymphoma, and in the legs, the most common location was the head and neck area (38%). These results may be due to the low number of patients in both the PCFCL (27) and PCDLBCL-LT (8) groups. However, the anatomical location has also been reported disparately in other studies with larger numbers of patients, and further debate is needed, especially in the PCDLBCL-LT subgroup, as to whether this nomenclature is the most appropriate for this aggressive subtype, as the different published series show that the location in the lower limbs is not always the most common location. Most likely, what is more important in the overall cohort of PCBCL patients is the appropriate histological classification rather than the location of the lesions. Another aspect to take into account with regard to the location of PCBLs is that they have a tropism for specific areas depending on the subtype, and this tropism is different when the cutaneous involvement is secondary to a lymph node lymphoma, which could also explain some discrepancies between the different published series [29]. In our study, multifocality was more common in the PCFCL subgroup (26%) than in the PCMZL and PCDLBCL-LT subgroups (19% and 12%, respectively). Although multifocality has been related to PCMZL, more recent retrospective studies have described more cases of multifocality in the PCFCL subgroup [35]. It is difficult to confirm whether these results were due to the retrospective nature of the study and the relatively small number of patients or whether there was a real tendency of PCFCL towards multifocality secondary to an improvement in histological classification.

For the treatments used, most patients received local treatments consisting of radiotherapy (55%) and surgery (50%). These two treatment strategies were also combined (21%) and systemic treatment (13%) in some patients. As noted above, local treatments are the cornerstone of PCBCL, and our study contrasts with the published data because the majority of these patients presented with localized disease amenable to either surgical excision or an acceptable radiotherapy field. Although the most commonly used dose in our patients was 36–40 Gy, the administration of low doses of RT (4 Gy) is recommended in multiple recently published studies because it does not decrease efficacy but rather improves toxicity, with very good cosmetic results that result in a better quality of life for patients, especially those with multifocal relapses [36]. In up to 37% of the patients, we were unable to collect radiotherapy doses because they were not included in the patients’ clinical history. This may be because, in many cases, radiotherapy is not administered at the same hospital where the patient was diagnosed, which may also explain the difficulty in recording all patient data in the years prior to the digitalization of medical records. With the implementation of electronic medical records, the quality of the data will likely improve significantly.

Systemic treatments alone or in combination with local treatments were administered to 62% of patients with PCDLBCL-LT and 22% of patients with PCMZL and PCFCL. The most commonly used option was R-CHOP followed by rituximab monotherapy; the latter was used for the indolent subtypes. In our series of cases, it was observed that in clinical practice, the treatments recommended in the clinical guidelines were mostly used [6]. Notably, the level of scientific evidence supporting treatment strategies for PCBCL is not greater than that for IVB since the existing data correspond to case series without randomized clinical trials due to the low overall incidence of these lymphomas. The wait-and-see strategy is also accepted for the two indolent subtypes, as occurs for nodal lymphomas. However, we did not object to this manoeuvre in these patients, probably because patients usually already require some type of treatment when they arrive at the oncologist’s office. A better understanding of these lymphomas at the molecular level is revealing a wide range of therapeutic possibilities that, in most cases, will be aimed at topical and local treatments. In those cases in which systemic treatment is necessary, new monoclonal or bispecific antibodies, CAR-T cells, ICIs, etc., could improve the prognosis of the most aggressive cases [37].

Moving to treatment efficacy, 96% of patients achieved CR after the first treatment (PCMZL 99%; PCFCL 93%; PCDLBCL-LT 88%). This finding confirms the efficacy of both local treatments, such as surgery and radiotherapy, which yield a response rate of more than 90%, and systemic treatments, such as chemoimmunotherapy, which have a response rate of more than 80%. However, since the data were collected from medical records and are retrospective, we could not objectively confirm whether complete responses were confirmed at the pathological level or only by clinical and radiological assessment of the patients. For this reason there could be biases in the type of responses, and in some cases incomplete resection or insufficient ablation could have led to some of the local recurrences recorded. In patients who achieved a CR after the first treatment, the median OS was not reached, whereas the median OS was 60 months in nonresponders. This highlights the importance of selecting a good initial treatment for patients with PCBCL. The combination of surgery plus radiotherapy for localized disease was associated with a significantly greater DFS than was the other combinations (*p* = 0.36) without affecting OS; therefore, it is unreasonable to administer the two treatments in the same therapeutic line since this combination can increase morbidity and worsen the cosmetic outcome in patients without a clear benefit unless clear involvement of the resection margins is detected in an excisional biopsy. In terms of recurrence, among the 45 recurrences, 22 were local (49%), 9 were regional (20%), 10 were distant (22%) and 4 were of an unknown location (9%). Patients with fewer than three local recurrences had better survival than did those with more than three recurrences (*p* = 0.023). Regarding overall survival (OS), 94.5% of the patients were still alive at 10 years (95.2% for PCMZL and PCFCL and 85.7% for PCDLBCL and LT). Patients with multifocal presentation had shorter PFS than did those with a single or regional location at diagnosis (*p* = 0.014); however, this difference did not seem to have an impact on overall survival. These results are consistent with the rest of the published series and highlight the favourable evolution of indolent PCBCL despite the various recurrences or multifocal presentations, as salvage treatments often work well. A separate case is the much more aggressive PCDLBCL-LT subgroup, which cannot be included in the same group. However, data are often reported in a common way due to the scarcity of cases in which overall survival is worse, although results ranging from 45% survival to 85–90% have been reported. In fact, the World Health Organization’s 2022 classification differentiates the indolent lymphoproliferative disorder subgroups (PCMZL, PCFCL, and EBV + MU) from patients with more aggressive subtypes, such as PCDLBCL-LT and IVLBCL, because of their distinct differences in management and prognosis [12]. In this subgroup, the emergence of new treatments for nodal cancer, especially refractory or relapsed disease, opens up a new range of treatment possibilities. However, the best way to determine the best treatment options for these patients is undoubtedly to better identify and classify them at the molecular level. It should be noted that, in this work, there was a very low number of patients with PCDLBCL-LT (n = 8), which may bias the survival results (OS 85.7% at 10 y). Therefore, these survival data should not be taken into account because of the small sample, the retrospective nature of the study, and the possibility of migration in the subtypes because the biopsies of the cases have not been reviewed for reclassification [38].

Finally, disease-specific survival (DSS) at 10 years was 97.2% in the whole cohort, which is also in agreement with the data known to date, reinforcing the idea that, globally, PCBCL has a very good prognosis [39].

## 5. Conclusions

PCBCL is a rare entity accounting for <2% of NHLs and has undergone heterogeneous classification, diagnosis, and therapeutic management in recent decades, with the PCMZL and PCFCL subgroups having an indolent course and the PCDLBCL-LT having an aggressive course. The various updates of the WHO classification since 2005 have contributed to a better grouping of these lymphomas with reclassification of some subtypes within others. The characteristics of the PCBCL in our group are comparable to those of other published series, although we found a higher frequency of PCMZL vs. PCFCL and a low frequency of leg localization of the PCDLBCL and LT (12%). There remains heterogeneity in therapeutic management, but according to published data and clinical guidelines, most cutaneous lymphomas can be treated with surgical excision or local radiotherapy with very good long-term results in patients with the least aggressive subtypes. Multifocality and recurrence tendency do not compromise the OS of patients with PCBCL, especially for indolent subtypes. Prospective studies based on the specific clinical and biological characteristics of each PCBCL subtype are needed to select treatments more appropriately without extrapolating from their nodal counterparts.

## Figures and Tables

**Figure 1 cancers-16-01034-f001:**
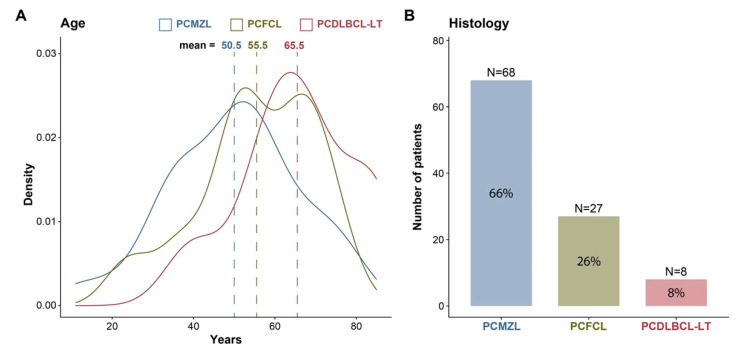
Clinical characteristics of the population. (**A**) Distribution and mean age according to histology; (**B**) number of patients according to their histology.

**Figure 2 cancers-16-01034-f002:**
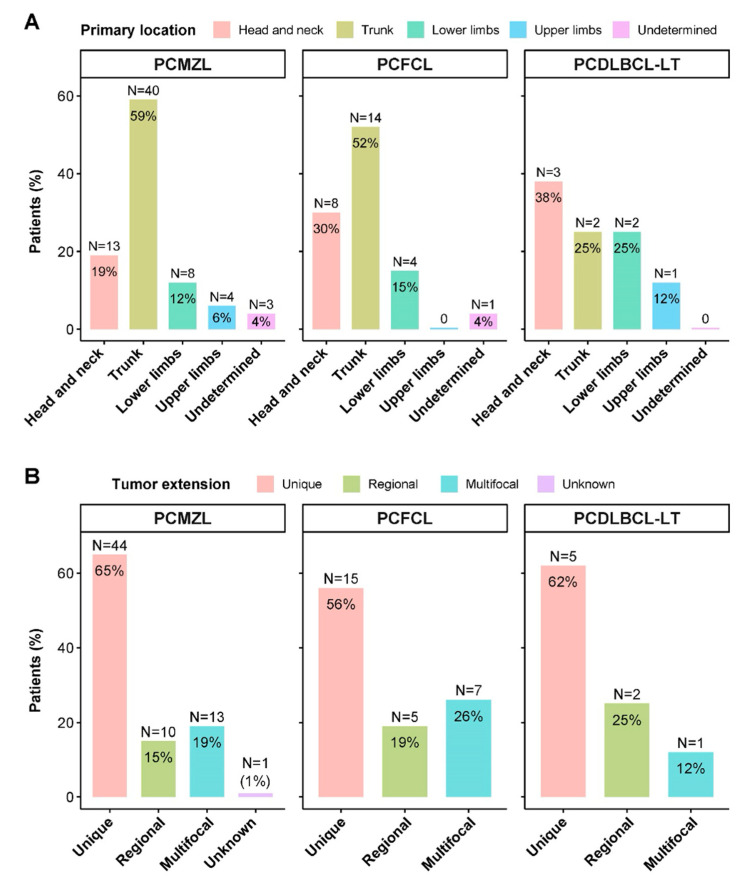
Tumour characteristics. (**A**) Tumour primary location stratified by histology; (**B**) tumour skin extension stratified by histology.

**Figure 3 cancers-16-01034-f003:**
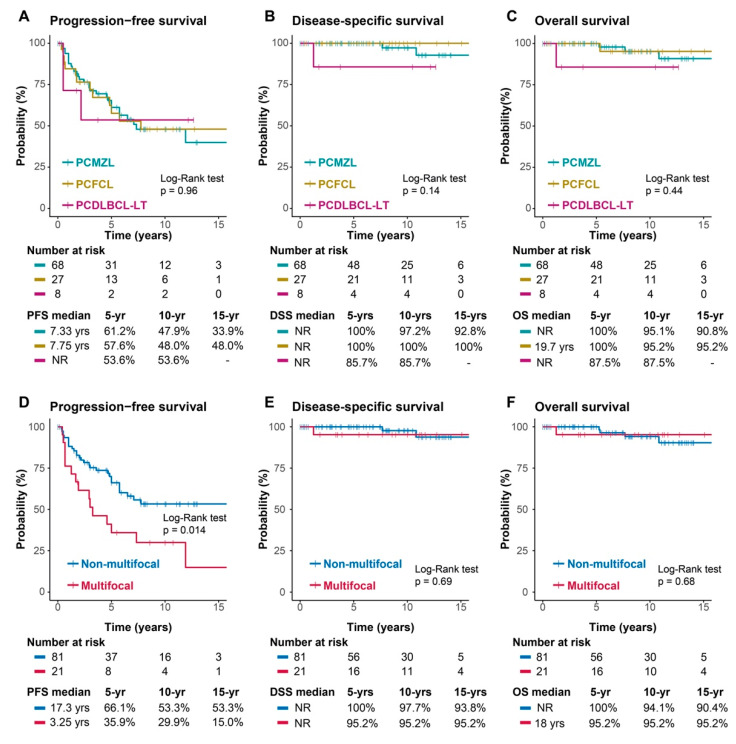
K‒M survival curves. The top row (**A**–**C**) is stratified by tumour histology, and the bottom row (**D**–**F**) is stratified by tumour extension. (**A**,**D**) progression-free survival; (**B**,**E**) disease-specific survival; (**C**,**F**) overall survival.

**Table 1 cancers-16-01034-t001:** Clinical characteristics.

Characteristics		Total Cases N = 103	PCMZL N = 68	PCFCL N = 27	PCDLBCL-LT N = 8
Gender	Female	53 (51%)	32 (47%)	18 (67%)	3 (38%)
	Male	50 (49%)	36 (53%)	9 (33%)	5 (62%)
Age	Median (range)	53 (11, 85)	51 (11, 82)	56 (24, 77)	66 (40, 85)
ECOG	0	90 (87%)	62 (91%)	23 (85%)	5 (56%)
	1	9 (9%)	5 (7%)	2 (7%)	2 (25%)
	2	1 (1%)	0	0	1 (12%)
	≥3	0	0	0	0
	Unknown	3 (3%)	1 (1.4%)	2 (7%)	0
Stage	I	79 (77%)	54 (79%)	20 (74%)	5 (62%)
	II	22 (21%)	13 (19%)	7 (26%)	2 (25%)
	III	0	0	0	0
	IV	2 (2%)	1 (1%)	0	1 (12%)
Prior skin disease	Yes	15 (14.6%)	12 (18%)	3 (11%)	0
	No	81 (78.6%)	50 (74%)	24 (89%)	7 (88%)
	Unknown	7 (6.8%)	6 (9%)	0	1 (12%)
Tumour main location	Head and neck	24 (23%)	13 (19%)	8 (30%)	3 (38%)
	Trunk	56 (54%)	40 (59%)	14 (52%)	2 (25%)
	Upper limbs	14 (14%)	8 (12%)	4 (15%)	2 (25%)
	Lower limbs	5 (5%)	4 (6%)	0	1 (12%)
	Undetermined	4 (4%)	3 (4%)	1 (4%)	0
Tumour extension	Unique	64 (62%)	44 (65%)	15 (56%)	5 (62%)
	Regional	17 (17%)	10 (15%)	5 (19%)	2 (25%)
	Multifocal	21 (20%)	13 (19%)	7 (26%)	1 (12%)
	Unknown	1 (1%)	1 (1%)	0	0
Primary outcome	Complete response	99 (96%)	67 (99%)	25 (93%)	7 (88%)
	No response	4 (4%)	1 (1%)	2 (7%)	1 (12%)
Relapse	Yes	45 (44%)	30 (44%)	12 (44%)	3 (38%)
Number of relapses	1–3 relapses	35 (34%)	24 (35%)	9 (33%)	2 (25%)
	>3 relapses	8 (7.8%)	5 (7.4%)	3 (11%)	0
	Unknown number	2 (1.9%)	1 (1.5%)	0	1 (13%)
Site of relapse	Local	22 (21%)	14 (21%)	6 (22%)	2 (25%)
	Regional	9 (8.7%)	8 (12%)	1 (3.7%)	0
	Distant	10 (9.7%)	6 (8.8%)	4 (15%)	0
	Unknown	4 (3.9%)	2 (2.9%)	1 (3.7%)	1 (13%)
Deaths	By lymphoma	3 (3%)	2 (3%)	0	1 (12%)
	By other causes	4 (4%)	2 (3%)	2 (7%)	0

Abbreviations: PCMZL = primary cutaneous marginal zone lymphoma; PCFCL = primary cutaneous follicle centre lymphoma; PCDLBCL-LT = primary cutaneous diffuse large B-cell lymphoma, leg type.

**Table 2 cancers-16-01034-t002:** Treatment combination.

Treatments		Total Cases N = 103	PCMZL N = 68	PCFCL N = 27	PCDLBCL-LT N = 8
Only local		73 (70.9%)	51 (75%)	19 (70.4%)	3 (37.5%)
	Only S	24 (23.3%)	19 (28%)	5 (18.5%)	0
	Only RT	27 (26.2%)	20 (29%)	7 (25.9%)	0
	S + RT	22 (21.4%)	12 (18%)	7 (25.9%)	3 (37.5%)
Only systemic		14 (13.6%)	9 (13.2%)	3 (11.1%)	2 (25%)
	Only CT or R-CT	10 (9.7%)	6 (8.8%)	2 (7.4%)	2 (25%)
	Only systemic rituximab	4 (3.9%)	3 (4.4%)	1 (3.7%)	0
Local + Systemic		12 (11.6%)	7 (8.8%)	3 (11.1%)	3 (37.5%)
	S + ST	5 (4.8%)	2 (2.9%)	1 (3.7%)	2 (25%)
	RT + ST	6 (5.8%)	3 (4.4%)	2 (7.4%)	1 (12.5%)
	S + RT + ST	1 (1%)	1 (1.5%)	0	0
Other therapies		4 (3.9%)	2 (3%)	1 (7.4%)	0
	RT + ILR	1 (1%)	1 (1.5%)	0	0
	Only ILR	1 (1%)	0	1 (3.7%)	0
	Only Intralesional corticoids	1 (1%)	0	1 (3.7%)	0
	Doxycycline	1 (1%)	1 (1.5%)	0	0

Abbreviations: S = surgery; RT = radiotherapy; CT = chemotherapy; R-CT = rituximab plus chemotherapy; ST = systemic treatment; ILR = intralesional rituximab.

**Table 3 cancers-16-01034-t003:** Treatment regimen.

Treatments		Total Cases N = 103	PCMZL N = 68	PCFCL N = 27	PCDLBCL-LT N = 8
Any surgery		52 (50%)	34 (49%)	13 (48%)	5 (62%)
Any CT		26 (25%)	15 (22%)	6 (22%)	5 (62%)
	CHOP	4 (15%)	2 (13%)	1 (17%)	1 (20%)
	R-CHOP	8 (31%)	4 (27%)	1 (17%)	3 (60%)
	R-CTX/R-CVP	3 (12%)	1 (7%)	1 (17%)	1 (20%)
	Rituximab	7 (27%)	4 (27%)	3 (50%)	0
	Cyclophosphamide	1 (4%)	1 (7%)	0	0
	Chlorambucil	2 (8%)	2 (13%)	0	0
	CT not specified	1 (4%)	1 (7%)	0	0
Any RT		57 (55%)	37 (54%)	16 (59%)	4 (50%)
	30–35 Gy	6 (11%)	5 (14%)	1 (6%)	0
	36–40 Gy	24 (42%)	19 (51%)	5 (31%)	0
	41–45 Gy	4 (7%)	2 (5%)	1 (6%)	1 (25%)
	46–50 Gy	2 (4%)	2 (5%)	0	0
	Dose not specified	21 (37%)	9 (24%)	9 (56%)	3 (75%)

Abbreviations: CHOP = cyclophosphamide, doxorubicin, vincristine, prednisolone; CVP = cyclophosphamide, vincristine, prednisolone; R = rituximab; CT = chemotherapy.

## Data Availability

Data are contained within the article and Appendix A.

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
