# Peer review of "A Descriptive Study of 103 Primary Cutaneous B-Cell Lymphomas: Clinical and Pathological Characteristics and Treatment from the Spanish Lymphoma Oncology Group (GOTEL)"

_cancers, 2024, doi:10.3390/cancers16051034_

Round 1

Reviewer 1 Report

Comments and Suggestions for Authors

The authors collected all cutaneous B-celll lymphomas from a 20 year period across several institutions. This is a difficult work. Due to the heterogeneity of this population the limitations of this study are significant. The concept is very interesting and provides real life data on this population of lymphomas.

However several coners have to be addresses to imporve the paper:

On line 282 the study population age is given, with 65 years of upper age. This is very low upper age, and later in the subtypes, indeed up to 74 years are mentioned. This has to be clarified.

Understanding the difficulty in collecting data from so many institutions, but more information should be given on the staging of the diseases. What was the perentage of PET/CT, CT in the groups ?

The DLBCL-LT group is very small, and all chaarcteristics are contradicting the literature. Only 1 patient is presented on the lower limg, the survival data is much better then predicted. This suggests that patients may be categorized into a wrong subgroup. group? I recommend that this subgroup's data is not presented in this detail, as the data may be scientifically misleading. These patients usually have much worse survival, relapses even with combinational therapy.

The discussion lists some limitations, but I recommend the limitations to be further expanded and also omit the conclusions on the LT group.

After these modifications it may be a very interesting paper for the readers.

Reviewer 2 Report

Comments and Suggestions for Authors

The authors provide a retrospective analysis of 103 patients with cutaneous non-Hodgkin lymphoma, who were variously treated with surgery, radiotherapy, chemotherapy, or a combination of these treatments.

Progression-free, disease-free, and overall survival were calculated.

Here are my observations:

Line 308-309: What about the remaining 1%? (62+20+17 = 99%)

The topic is focused on the effects of local and systemic therapies. Indeed, recurrence and disease-free survival are investigated, but there is no information or data about the pathological confirmation of radical excision or complete ablation of such lesions. Could incomplete resection or ablation cause some local recurrences?

Was the surgery performed only once per each individual lesion or did some of them undergo re-excision to achieve free margins?

In how many cases of a combination of surgery and radiotherapy, the latter was used to sterilize a cancer that was not completely removed by the former? In how many cases radiotherapy was used exclusively with adjuvant intent?

How was the complete response determined, only by clinical observation? Or due to the absence of a medical report?

The follow-up was conducted by different physicians, right? Which was the scheduled follow-up procedure, clinical or not? Any instrumental investigation? For which patients? Which time interval with?

What was the rate of adhesion among patients who were followed up? The statistics could be greatly influenced by it.

In summary, the manuscript is overall well-written, the topic is interesting, the results are clearly presented, the discussion is balanced, and shows the limitations of such a type of study.

But on the other hand, the data origin is uncertain, as with any retrospective design study, and the follow-up was conducted in an undefined manner, so definitive conclusions could not be drawn based on these findings.

These biases affect the overall merit of the work, and I believe they cannot be corrected since the data source was uncontrolled.

Round 2

Reviewer 1 Report

Comments and Suggestions for Authors

Thanks you for the revision. With these added information the paper is much better, and now acceptable.